# Atmospheric Charge Separation Mechanism Due to Gas Release from the Crust before an Earthquake

**Wen Li** [1,2], **Zhibin Sun** [1,*], **Tao Chen** [1,*], **Zhaoai Yan** [1], **Zhongsong Ma** [3], **Chunlin Cai** [1], **Zhaohai He** [1], **Jing Luo** [1] **and Shihan Wang** [1,2]

1   State Key Laboratory of Space Weather, National Space Science Center, Chinese Academy of Sciences, Beijing 100190, China; liwen@nssc.ac.cn (W.L.); yanza@nssc.ac.cn (Z.Y.); clcai@nssc.ac.cn (C.C.); he_zh@nssc.ac.cn (Z.H.); jluo@spaceweather.ac.cn (J.L.); xutwsh3@126.com (S.W.)
2   College of Earth and Planetary Sciences, University of Chinese Academy of Sciences, Beijing 100049, China
3   Technology and Engineering Center for Space Utilization, Chinese Academy of Sciences, Beijing 100094, China; mazhongsong@csu.ac.cn
*   Correspondence: zbsun@nssc.ac.cn (Z.S.); tchen@nssc.ac.cn (T.C.); Tel.: +86-138-1019-5052 (Z.S.); +86-139-1042-1558 (T.C.)

**Abstract:** In fair weather, the vertical atmospheric electric field is oriented downward (positive in the earth surface ordinate system) in the global atmospheric circuit. Some researchers have revealed the unique phenomenon whereby once an upward vertical atmospheric electric field is observed in fair weather, an earthquake (EQ) follows within 2–48 h regardless of the EQ magnitude. However, the mechanism has not been explained with a suitable physical model. In this paper, a physical model is presented considering four types of forces acting on charged particles in the air. It is demonstrated that the heavier positive ions and lighter negative ions are rapidly separated. Finally, a reversed fair weather electrostatic field is formed by the above charge separation process. The simulation results have instructive significance for future observations and hazard predictions and still need further research.

**Keywords:** charge separation; reverse vertical atmospheric electric field; multiple forces

## 1. Introduction

The concept of the global atmospheric circuit (GAC) is essential for the study of atmospheric electricity. The atmospheric electric field is particularly important in the conceptual modeling process of the GAC and is an important characteristic parameter in the fields of space and atmospheric physics. The GAC and related influencing factors are shown in Figure 1, and it is revealed that the Earth's ionosphere and the solid Earth are favorable conductors. All positive and negative charges are distributed only across the surface of these conductors. On the one hand, many positive charges are located on the surface of the ionospheric bottom. On the other hand, many negative charges are located on the Earth's surface. Globally, 2000 lightning strikes per second provide a positive charge to the ionospheric bottom and may generate a 2000 A current at the surface of the ionospheric conductor, i.e., a partly plasma-induced current. The ionosphere contains the positive electrode of the air–ground capacity, while the ground provides the negative electrode of the air–ground capacity. In fair weather, the current flows to the ground surface from the bottom of the ionosphere (at an altitude of approximately 60 km). Therefore, the daily curve of the atmospheric electrostatic field shows an absolute positive value around 100 V/m (at the lower right corner of Figure 1, and it is the daily variation curve of the atmospheric electric field observed in Beijing, which is just an example, the horizontal axis is the UT time, and the vertical axis indicates the value of atmospheric electric field with V/m) [1–6]. Meteorological activities are closely related to the atmospheric electric field. Notably, galactic cosmic rays, high-energy particle precipitation and coronal mass ejection

can affect the atmospheric electrostatic field by altering the air conductivity or ionospheric potential. Underground radioactive gas release can also affect the near-surface atmospheric electric field.

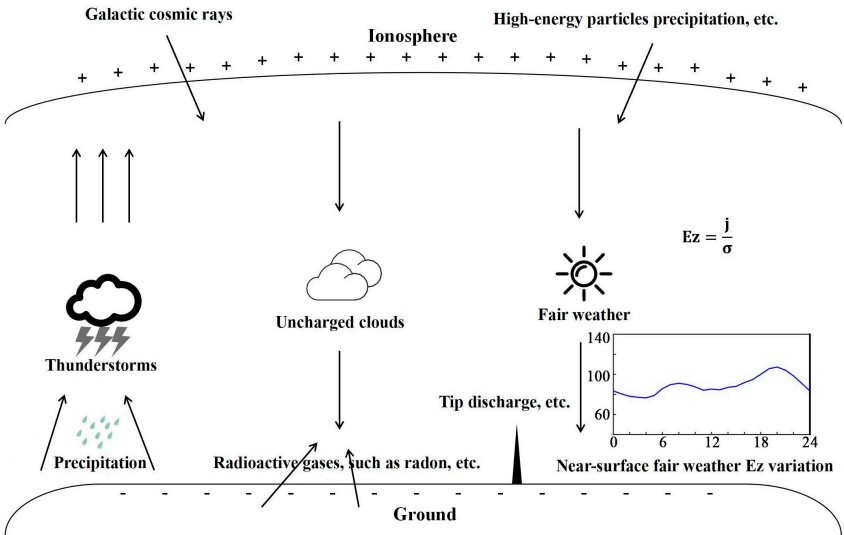

**Figure 1.** Background of the global electric circuit ("+", "-" indicate that Earth's surface carries negative charges and ionosphere carries positive charges. $E_z$, j and σ denote the atmospheric electric field, air–earth current and conductivity along the vertical direction, respectively. The bottom-right corner is an example of the daily variation of atmospheric electric field.).

There are four types of emissions from the crust into the atmosphere before an impending EQ: electromagnetic, acoustic, thermal and ionized emissions [7–9]. The spatial and temporal scales of these four emissions differ in geospace. In general, precursor signals of electromagnetic emissions emerge in geospace 1 to 40 days before an EQ [10,11], precursor signals of acoustic emissions emerge in geospace 1 to 30 days before an EQ and precursor signals of thermal emissions emerge in geospace 1 to 30 days before an EQ [12]. Atmospheric electrostatic field monitoring has recently become increasingly popular and has been widely studied [13–20]. Precursor signals of ionized emissions emerge in geospace 2 to 48 days before an EQ. Atmospheric electrostatic field anomalies prior to EQs have been widely studied. Since the Tangshan EQ in 1976, China has established several atmospheric electrostatic field monitoring stations, and obvious anomalous cases have been observed [14]. Omori et al. [20] noted that anomalous radon emissions trigger considerable decreases in the atmospheric field of the lower atmosphere (from the ground to an altitude of 2 km), as observed around the time of the Kobe EQ in 1995. Omori et al. [20] further suggested that the behavior of radon in terms of the atmospheric electrostatic process could explain seismic precursors observed near the ground. Choudhury et al. [13] described the characteristics of the vertical atmospheric electrostatic field as negative 7–12 h before an EQ according to the statistics of 30 EQ events of various classes across northern India. Smirnov [18] reported that more than one hundred cases showed negative $E_z$ anomalies approximately one day in advance of Ms 4–6 EQs, but there was no obvious relationship between the $E_z$ value and the epicentral location, similar to that between $E_z$ and the magnitude. When large amounts of gases are released from the crust into the atmosphere, radioactive elements such as $^{222}$Rn and $^{214}$Th in these gases undergo alpha, beta and gamma decay processes. For example, an alpha particle contains 4.53 MeV of energy, which is sufficient to ionize 1,500,000 gas molecules, because an atmospheric molecule only needs 32 eV of energy for ionization. Therefore, large amounts of positive and negative ions are produced in the atmosphere by radioactive matter originating from the crust. Finally, a unique polarized electric field can be formed in the atmosphere that can alter the direction of the formal background atmospheric vertical electric field [21], as shown in Figure 2. Four

cases (the Beijing Ms 3.0, Rongxian Ms 4.7, Changning Ms 6.0, and Wenchuan Ms 8.0 EQs) of preseismic atmospheric electric field hourly scale anomalies are shown in Figure 2, and the time scales for the advancement of the atmospheric electrostatic anomalies are 3.8, 11, 23.5 and 7 h, respectively.

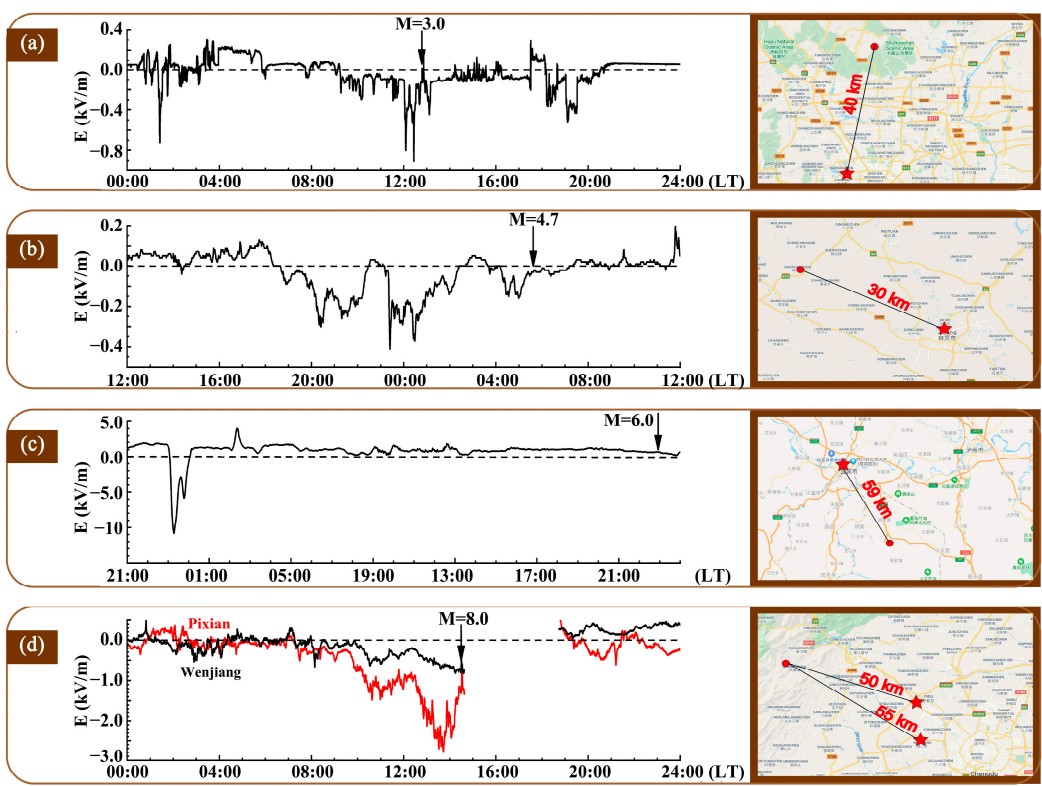

**Figure 2.** Abnormal signals of atmospheric static electricity on sunny days basically emerge only 2–48 h before an earthquake (panel (**a**) shows the Beijing Ms 3.0 earthquake on 14 April 2019, with a station located 40 km from the epicenter, (**b**) shows the Rongxian Ms 4.7 earthquake on 24 February 2019, with a station located 30 km from the epicenter, panel (**c**) shows the Changning Ms 6.0 earthquake on 17 June 2019, with a station located 50 km from the epicenter and (**d**) shows the Wenchuan Ms 8.0 earthquake on 12 May 2008, with two stations located 50 and 55 km from the epicenter. The data gap in panel (**d**) is due to a power outage. In the maps on the right, the stars denote the stations and the circles denote the epicenters) [21].

How are hourly scale atmospheric electric field anomalies generated before earthquakes? What forces actually cause atmospheric charges to separate, and how are these positive and negative charges separated? The separation of positive and negative charges in the atmosphere is essential for the physical mechanism of preseismic atmospheric electric field formation. In this paper, we present simulation experiments conducted based on these queries.

## 2. Physical Modeling

Earth serves as a significant source for the generation of numerous gases which are byproducts of natural radioactive elements such as uranium, thorium and potassium. These gases provide crucial insights into the Earth's interior, the conditions of the massifs in the upper crust and the geoecological environment. More than 60–70% of the general productivity of natural ionization sources is provided by ground radiation. Through ground-based measurements, gaseous discharges of various substances, including radioactive elements such as radon and its decay products, have been detected in the atmosphere prior to the occurrence of major EQs. The concentration of these gases experiences a substantial increase of approximately 4–8 times over undisturbed levels preceding an EQ. A rise in the radon

concentration can be observed several days before the onset of seismic activity, reaching emission levels as high as 25 Bq/m$^3$, which correspond to an ion formation velocity of approximately Q~16 × 10$^3$ cm$^{-3}$ s$^{-1}$. These variations persist for periods ranging from several hours to several days. Within the Earth's surface, one can identify three primary ionization sources, namely, alpha, beta and gamma radiation. However, at a distance of 1–2 m above the surface, the average intensity of ion production resulting from these terrestrial sources does not exceed 3–5 cm$^{-3}$ s$^{-1}$. Notably, the radioactivity present in the air contributes to the intensity of ion production within the lower layer of the atmosphere, accounting for approximately 3–4 cm$^{-3}$ s$^{-1}$ [22].

As shown in Figure 3, there are three main types of ionization in the tropospheric atmosphere: radioactive radiation in the Earth's crust ($\gamma$ rays constitute the main ionization source), radioactive radiation in the atmosphere ($\alpha$ rays constitute the main ionization source) and cosmic rays from space. The verse lines are high-energy particles from space, mainly encompassing a 10$^8$–10$^{20}$ eV high-energy proton composition. Energetic GeV particles can penetrate the atmosphere to reach the ground, which can impact the molecular atoms in the atmosphere to form high-energy particles, referred to as secondary cosmic lines. Due to the geomagnetic field, the cosmic line is deflected toward the poles when it penetrates the Earth. Therefore, the intensity increases with latitude. The total atmospheric ionization rate at the ground is 9.8 cm$^{-3}$ s$^{-1}$ and the radioactivity in the atmosphere accounts for 47% of the total radioactivity, which in the crust accounts for 36% of the total radioactivity, and that of the cosmic line accounts for 17% of the total radioactivity. This suggests that at the surface, the radioactive material in the atmosphere plays a major role in atmospheric ionization [22].

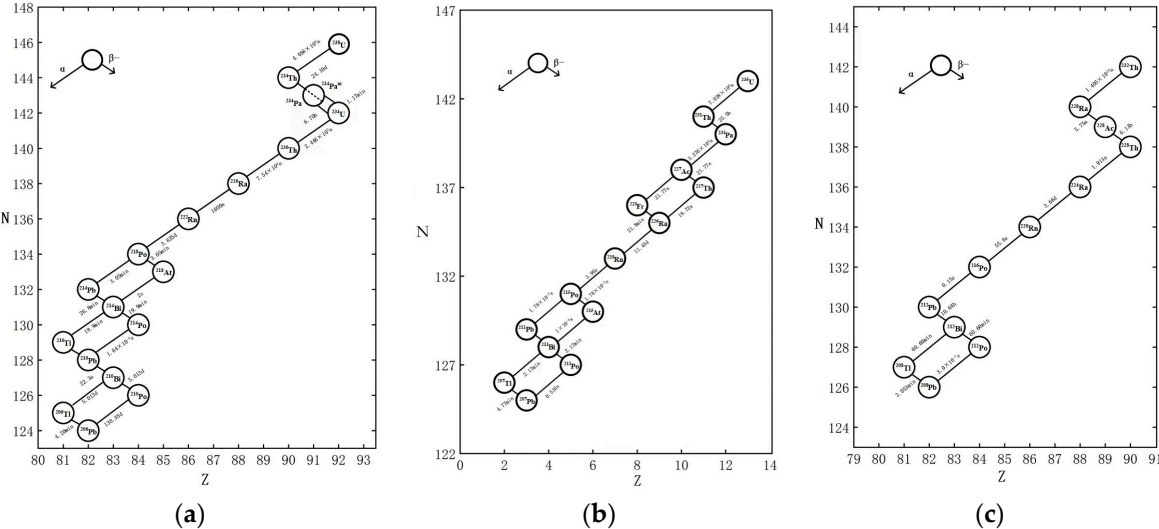

**(a)**      **(b)**      **(c)**

**Figure 3.** Decay of radioactive elements and their decay periods (Units of time: a—year, d—day, h—hour, and min—minute, s—second. (**a**) shows the uranyl system decay, (**b**) shows the actinide system decay and (**c**) shows the thorium system decay. The upper left corner indicates the two directions for a-decay and b-decay, respectively.).

Notably, $\alpha$-ray penetration is very poor, ranging from 1 to 8 cm, with atmospheric ionization rates ranging from 40 to 1 cm$^{-3}$·s$^{-1}$. The range of the ionization effect of radioactive materials in the Earth's crust on the atmosphere is limited to less than 1 km, the influence range of the radioactive material in the atmosphere is limited to less than 5 km and the cosmic line increases with increasing height. In the 3–4 km range, cosmic rays account for 97% of the total radioactivity, and within the range of 5–6 km and above, the atmospheric ionization rate is largely produced by cosmic rays. Whether over land or sea, 3 km is the dominant cosmic line, so the above still applies.

Planetary geological activities can significantly impact the ionization environment of space. Planetary geological activities can cause anomalous signals in the atmospheric electric field, and the possible mechanism by which planetary geological activities can alter the ionization environment of space was analyzed and determined. The movement of solid planets, such as the Earth, moon, Mars and other planets, leads to the fragmentation or melting of rocks and the dissolution or phase change of minerals. Some of the daughter isotopes of radioactive parent isotopes retained in certain minerals or rocks can be released in large quantities. In this decay process, many particles are also released, with a particle energy of 5.2 MeV. Atmospheric analysis requires ionization to generate 32 eV of energy, so one particle can produce 150,000 pairs of positive and negative particles, thus filling the air with numerous ion pairs.

At the stage of planetary geological activity, the ionizing radiation of radioactive material produces millions or even hundreds of millions of ion pairs, which promotes the hydration process in the atmosphere (water vapor is combined with other particles). In the hydration process, the excess kinetic energy of molecules is released, which is then converted into heat energy, resulting in the characteristic heat release. Measurement of the radioactive material in planetary environments can facilitate a greater understanding of future conditions.

Radioactive decay involves four main processes: the natural uranium, actinium and thorium series, plus an artificial radiation system. The uranium series, starting with $^{238}$U and ending with sTable $2^{06}$Pb, includes notable isotopes like $^{222}$Rn and $^{234}$Th, with 238U having a half-life of 4.468 billion years. The actinium series begins with 235U, ending with sTable $2^{07}$Ph, and involves intermediates such as 227Ac (half-life of 21.772 years) and 223Ra. The thorium series, starting with 232Th and ending with sTable $2^{08}$Ph, includes isotopes like $^{228}$Ra and $^{220}$Rn, with $^{232}$Th having a half-life of 14 billion years. These series release substantial energy, contributing to underground heat and potentially influencing seismic activity. The high density of isotopes like $^{238}$U, with a density of 19.1 g/cm$^3$, is significant for industrial applications and cosmic dating.

Water molecules are combined more easily with positive ions. Based on hydration processes and atmospheric sources of positive and negative ion production [23], we can assume that the mass of positive ions is greater than that of negative ions. To simulate the main physical process, only a single positively charged particle and a single negatively charged particle were considered. All suitable forces eventually result in the rapid separation between two particles with different charges. Logically, numerous positive and negative ions undergo a separation process. Finally, the separated ions form a corresponding polarized electric field in the air. In general, the corresponding polarized electric field is oriented upward (from the ground to the sky).

Notably, radon, thoron and other radioactive elements released underground continue to release energy as they decay into α, β and γ particles, and the decay process continuously occurs underground or aboveground. Then, the process of α particle decay that promotes the ionization of molecules in the atmosphere also causes heat release. In the continuous microfracturing process of the Earth's crust, heat from the underground is released into the near-surface atmosphere. In addition, coupled with the consideration of longwave radiation anomalies prior to earthquakes [7], there must be a thermal driver *F* before the earthquake, but the variation in the F value is complex. Therefore, here, *F* is described only qualitatively.

In general, a charged particle in the air experiences four forces: atmospheric quasistable electrostatic force qE, gravity force mg, air drag force $kv^2$ and thermal convection (upward thermal pressure) force *F*, where *q* is the ion charge, E is the average atmospheric electrostatic field, m is the ion mass, g is the gravitational acceleration, *k* is the drag coefficient and *v* is the ion velocity.

Single positive and negative ion particles are considered in the following equations:

A single atmospheric positive ion satisfies the following equation:

$$m_+ \frac{dv_+}{dt} = -q_+ E - m_+ g - k v_+^2 + F \tag{1}$$

A single atmospheric negative ion satisfies the following:

$$m_- \frac{dv_-}{dt} = q_- E - m_- g - k v_-^2 + F \tag{2}$$

Then, the average atmospheric electrostatic field E can be obtained as:

$$E = E_0 e^{-\alpha z} \tag{3}$$

where $m_+$ and $m_-$ are positive and negative ion masses, respectively, $q_+$ and $q_-$ are the amounts of positive and negative ion charges, respectively, $g$ is the gravitational acceleration, $k$ is the air drag coefficient and $F$ is the upward thermal driving force. The above equations constitute a simplified fundamental model, where $F$ is regarded as a constant and $v_+$ and $v_-$ are the velocities of positive and negative ions, respectively. Accounting for the actual measured distributions of the atmospheric electric field with altitude, we considered the background electric field E here as a quantity that only varies with height; it varies vertically downward under normal fair-weather conditions, and the change in height follows an exponential distribution [24]:

The parameters in the quantitative model and their physical explanations are as follows:

$m_+$: $1.63 \times 10^{-17}$ kg (the mass of positive ions including many molecules);

$m_-$: $4.08 \times 10^{-18}$ kg (the mass of negative ions including many molecules);

$g$: 9.8 m/s$^2$ (gravitational acceleration);

$q_+$: $1.6 \times 10^{-18}$ C (the charge of positive ions including many molecules);

$q_-$: $1.6 \times 10^{-18}$ C (the change in positive ions including many molecules);

$k$: 0.45 (drag coefficient);

$E_0$: 100 V/m (average value of the atmospheric electrostatic field under fair weather conditions);

$a$: $1.7 \times 10^{-5}$/m.

Assuming that F is an exponentially decreasing quantity with time, substituting Equation (3) into Equations (1) and (2), with some constant values set as described above, leads to the simulation results, and the results can be obtained within 300 s, as shown in Figure 4.

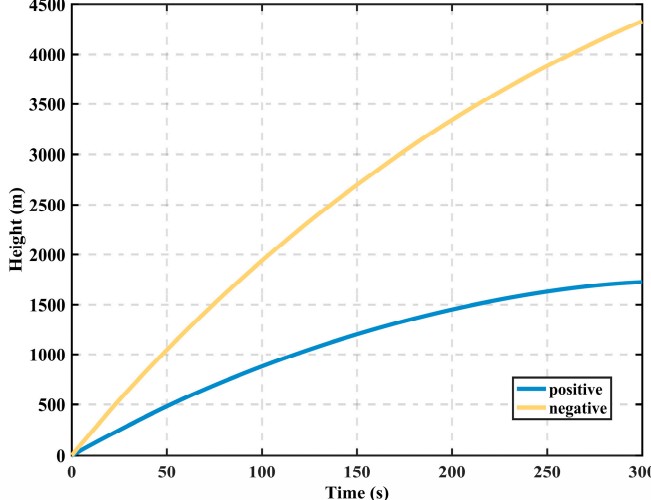

**Figure 4.** Simulation results illustrating the atmospheric charged particle process in which positive and negative ions are rapidly separated over time (blue and yellow colors indicate the movement of positive and negative ions, respectively.).

### 3. Simulation Results

Experimental evidence has revealed that the mobility of negative ions is approximately 1.3–1.4 times higher than that of positive ions. This disparity in mobility can likely be attributed to the asymmetry in the energy needed to remove ions of opposite charges from oxygen atoms in water molecules. Under the combined action of the electric field force, gravity, air resistance and thermal driving force, the upward movement speed of negative ions is higher than that of positive ions, and the two form an upward electrostatic field. When the charge further increases, the background field is reversed; notably, the so-called negative anomaly of the atmospheric electrostatic field occurs under fair weather conditions.

As shown in Figure 5, when subjected to the natural atmospheric electric field (E), positive ions experience a downward movement toward the Earth's surface, where they eventually recombine. However, due to their relatively low mobility, a spatial layer of positive ions is formed near the surface over time. In contrast, negative ions exhibit vertical upward movement (neglecting the electrons within the scope of this model due to their low concentration at the Earth's surface). Consequently, a near-ground electrode layer is established, characterized by a local electric field (El), which causes the natural atmospheric electric field to become negative. The primary consequence of the electrode effect is the development of an uncompensated electric charge across the ground surface. Turbulent movements and regular winds can transport this charge within the near-ground atmosphere, leading to the creation of an anomalous electrode layer over extensive areas. However, in reality, such situations typically persist for a short period since further turbulent movements tend to disrupt the electrode layer, mixing all ions and restoring electrical equilibrium. This physical process confirms the preseismic atmospheric electric field hourly scale anomalous signal shown in Figure 2.

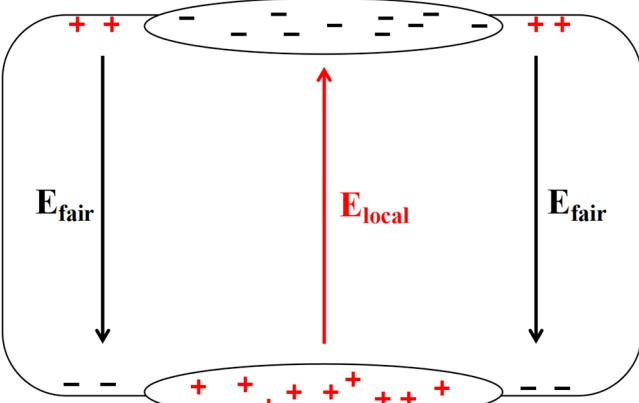

**Figure 5.** Schematic illustration of the movement of positive and negative charges to form a localized anomalous electric field ($E_{local}$) that is reversed from fair-weather atmospheric electric field ($E_{fair}$) ("+" and "−" are consistent with the interpretation of Figure 1).

### 4. Discussion

Based on the simulation results, positive and negative ions are mainly driven by four forces: positive and negative ions are advancing but are rapidly separated, positive ions form a charged layer and negative ions form a negative charge layer. These charges form an opposite electric field in the atmosphere near the surface under clear weather conditions, which is the opposite of the electric field under clear weather conditions depicted in Figure 1, as shown in Figure 6.

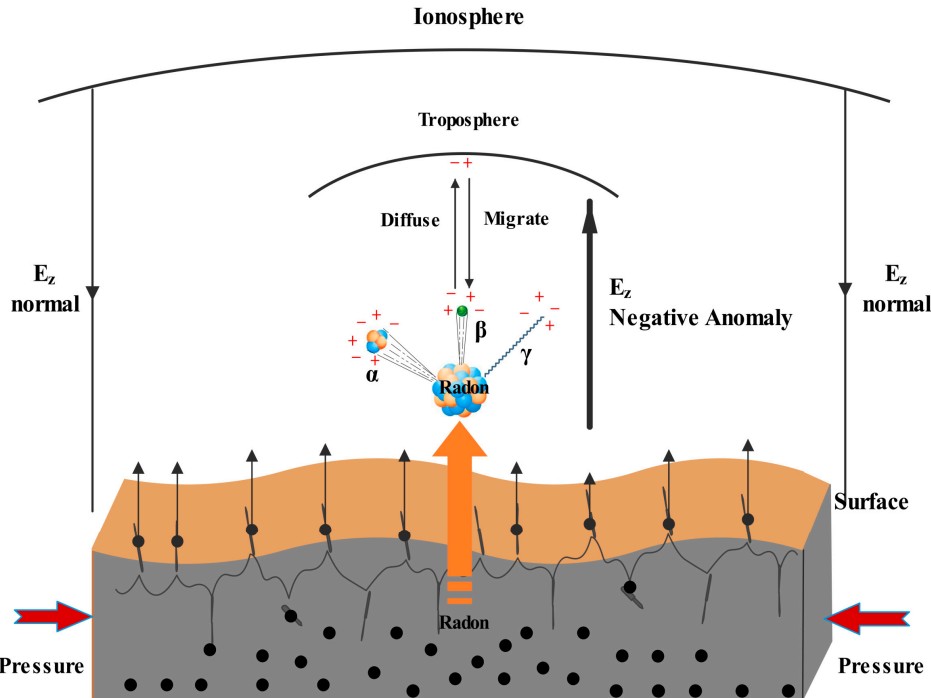

**Figure 6.** A reverse electrostatic field appears after atmospheric charges are separated due to gases stemming from the crust before an earthquake [25]. ("+", "−" indicate positive and negative charges).

The presented model is only a simplified one. In reality, collisions between positive and negative ions and electrostatic field forces must be accounted for, and the thermal driving force and drag coefficient also vary. Furthermore, due to the chemical reactions between positive and negative ions and between them and water molecules, their compositions change, and this process causes changes in their charge and mass. The proposed model is only one possibility for explaining the negative atmospheric electric field anomalies observed before EQs, but the physical processes before EQs (anomalies in the electron density of the ionosphere, release of radioactive gases, etc.), as well as the coupling relationships, require further in-depth study.

## 5. Conclusions

Based on some fundamental assumptions and physical formulas, we have simulated the process involving the generation of a large number of positive and negative ions by radioactive gases before an earthquake. The simulation results suggest that, under ideal conditions, positive and negative ions can be separated due to the combined influence of gravity, atmospheric electrostatic force, thermal driving force and air resistance. This separation could lead to the formation of a negative anomaly in the near-surface atmospheric electric field. However, it is important to note that these are merely idealized simulation results, indicating only a possibility. Further in-depth research is required to validate these findings.

Our simulation results could be used to better understand the ionization of crustal matter into the atmosphere, the vertical migration of charged particles and ultimately the formation of vertical electrical separation in air, resulting in an additional vertical polarization electric field. This could lay the foundation for the analysis of mechanics. Finally, it was observed that under the combined action of atmospheric static electricity, gravity, air resistance and upward thermal pressure, seed ions are provided by ionizing radiation of radioactive substances before an EQ. In addition to the different types and weights of positively and negatively charged particles, it is entirely possible for new positive and negative ions to undergo electrical separation in several minutes.

Based on the assumption that horizontal direction movement is neglected, the rapid separation mechanism of a pair of differently charged particles with different masses in the air depends on four forces: atmospheric formal vertical downward quasi-stable electrostatic force, vertical downward gravity force, air drag force (reverse to the direction of ion movement) and thermal convection (vertical upward thermal pressure) force F. It could be logically concluded that after charges are separated, a corresponding macroscopic polarized electrostatic field is rapidly established in the atmosphere. The resultant polarized electrostatic field is oriented toward the sky and exhibits the opposite direction to that of the daily atmospheric electrostatic field (vertical downward, as shown in Figure 4). This phenomenon is particularly obvious under fair weather conditions. Therefore, the charge separation mechanism could facilitate future EQ prediction.

**Author Contributions:** T.C. conceptualized this study. Z.S. and W.L. processed and analyzed the model and prepared the original draft, with contributions from all authors. Z.Y., Z.M., Z.S., C.C., Z.H., J.L. and S.W. contributed to the discussion. All authors have read and agreed to the published version of the manuscript.

**Funding:** This research was funded by the Scientific Instrument Developing Project of the Chinese Academy of Sciences (Grant No. YJKYYQ20190008), the National Key R&D Programme of China under grants 2018YFA0404201 and 2018YFA0404202, the Strategic Pioneer Program on Space Science, the Chinese Academy of Sciences (Grant Nos. XDA15016300, XDA15350201, and XDA17010303), the Pandeng Program of the National Space Science Center, the Chinese Academy of Sciences and the Ground-Based Space Environment Monitoring Network (Meridian Project II).

**Institutional Review Board Statement:** Not applicable.

**Informed Consent Statement:** Not applicable.

**Data Availability Statement:** The source code used in the model can be found on the github site: https://github.com/leibupang123/particlemove (accessed on 27 December 2023).

**Conflicts of Interest:** The authors have no financial interest to disclose.

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
