# Peer review of "Atmospheric Charge Separation Mechanism Due to Gas Release from the Crust before an Earthquake"

_applsci, doi:10.3390/app14010245_

Round 1

Reviewer 1 Report (Previous Reviewer 1)

Comments and Suggestions for Authors

The authors revised the text and accepted the reviewers' suggestions for improvement. The text could be accepted. On the scientific side, I have no objections. But first of all, technical deficiencies should be corrected.

It is unusual to place references in the abstract. This should be corrected. In the conclusions and abstract, it should be emphasized that these are simulation results and that more research is needed. The text on axes of Figures 2 and 3 should be enlarged. Also, authors should check all references and remove the remains of some text.

Author Response

#Reviewer 1

The authors revised the text and accepted the reviewers' suggestions for improvement. The text could be accepted. On the scientific side, I have no objections. But first of all, technical deficiencies should be corrected.

It is unusual to place references in the abstract. This should be corrected. In the conclusions and abstract, it should be emphasized that these are simulation results and that more research is needed. The text on axes of Figures 2 and 3 should be enlarged. Also, authors should check all references and remove the remains of some text.

Response: [Thank you for your valuable suggestions. The placement of references in the abstract has been corrected as you suggested. Then we have added in the conclusions and abstract that the findings are based on simulation results and have acknowledged the need for further research. And the text size on the axes of Figures 2 and 3 has been increased for better readability. Finally, we have checked all references and remove the remains of some text. I appreciate your attention to detail again.]

We tried our best to improve the manuscript and made some changes in the manuscript. We earnestly appreciate the Editors/Reviewers’ work and hope that the corrections will be met with approval. Once again, thank you very much for your comments and suggestions.

Reviewer 2 Report (Previous Reviewer 2)

Comments and Suggestions for Authors

I suggest the authors to significantly improve the conclusion highlighting better the results, and also describing more in detail the experimental part.

I also suggest to move up the chapter discussion prior of the chapter conclusion, revising a little bit the text of the two ones.

Line 110 I do not understand what "eman" means, please clarify.

Lines 118-142 justify the assumptions with proper references.

Lines 165-224 I suggest to simplify this part reducing the details.

FIGURES the captions have to be self-explanatory, I suggest to revise them accordingly (e.g., figure 3).

FIG.1 not clear the graph at bottom right, enlarge and explain

FIG.2 not clear enlarge and explain

Comments on the Quality of English Language

The English is sufficiently correct but we can suggest to try to be simpler and more fluent

Author Response

#Reviewer 2

I suggest the authors to significantly improve the conclusion highlighting better the results, and also describing more in detail the experimental part.

Response: [We really appreciate your useful comments, and the conclusion and experimental part have been revised.]

I also suggest to move up the chapter discussion prior of the chapter conclusion, revising a little bit the text of the two ones.

Response: [Your suggestion to move the discussion chapter before the conclusion and revise the text has been implemented. Thank you for this helpful recommendation.]

Line 110 I do not understand what "eman" means, please clarify.

Response: [Thank you for your guidance. It means radioactive gas concentration and is the same as Bq/m3. To make it more understandable, we have replaced it with Bq/m3.]

Lines 118-142 justify the assumptions with proper references.

Response: [Thank you, the reference has been provided in the revised version.]

Lines 165-224 I suggest to simplify this part reducing the details.

Response: [This section has been simplified to reduce excessive details, as you suggested. Thank you for pointing out the need for this.]

FIGURES the captions have to be self-explanatory, I suggest to revise them accordingly (e.g., figure 3).

Response: [All figure captions, especially for Figure 3, have been revised.]

FIG.1 not clear the graph at bottom right, enlarge and explain

FIG.2 not clear enlarge and explain

Response: [The graph in Figure 1 (bottom right) and in Figure 2 have been enlarged and explained. Thank you for your advice on making these figures more understandable.]

We tried our best to improve the manuscript and made some changes in the manuscript. We earnestly appreciate the Editors/Reviewers’ work and hope that the corrections will be met with approval. Once again, thank you very much for your comments and suggestions.

Reviewer 3 Report (Previous Reviewer 3)

Comments and Suggestions for Authors

This article describes the physical basis of a method for monitoring charge separation in the atmosphere before an earthquake. The purpose of studying these changes is to use them as an earthquake pre-indicator. One of the positive points of this research is that it is helpful in seismic-pron areas to reduce the seismic risk. Although the article has been improved compared to the previously submitted version, it needs major revision before being accepted for publication.

Comments:

1. Figure 2: The quality of Figure 2 is inadequate and requires improvement.

2. Figure 4: Please determine which of the relationships introduced in the article is used for extracting these graphs.

3. Conclusion: The conclusion section should be the last part of the article, and it should be without figures. Please remove Figure 6 from this section and move it to the discussion section. In addition, section 5 should be moved to the end of section 3 or added to the conclusion section. In any case, the summary and conclusion section should be the last topic.

Comments on the Quality of English Language

Minor editing of the English language is required.

Author Response

#Reviewer 3

This article describes the physical basis of a method for monitoring charge separation in the atmosphere before an earthquake. The purpose of studying these changes is to use them as an earthquake pre-indicator. One of the positive points of this research is that it is helpful in seismic-pron areas to reduce the seismic risk. Although the article has been improved compared to the previously submitted version, it needs major revision before being accepted for publication.

Comments:

  1. Figure 2: The quality of Figure 2 is inadequate and requires improvement.

Response: [The quality of Figure 2 has been improved as suggested. Thank you for bringing this to my attention.]

  1. Figure 4: Please determine which of the relationships introduced in the article is used for extracting these graphs.

Response: [We really appreciate your useful comments, substituting Eq. 3 into Eq. 1 and Eq. 2, with some constant values set as described in the text, leads to the results in Fig. 4, which are also explained in the revised manuscript.]

  1. Conclusion: The conclusion section should be the last part of the article, and it should be without figures. Please remove Figure 6 from this section and move it to the discussion section. In addition, section 5 should be moved to the end of section 3 or added to the conclusion section. In any case, the summary and conclusion section should be the last topic.

Response: [Thank you, the Figure 6 has been moved from the conclusion to the discussion section. The discussion has been moved to precede the conclusion section.]

We tried our best to improve the manuscript and made some changes in the manuscript. We earnestly appreciate the Editors/Reviewers’ work and hope that the corrections will be met with approval. Once again, thank you very much for your comments and suggestions.

Round 2

Reviewer 3 Report (Previous Reviewer 3)

Comments and Suggestions for Authors

The authors have made revisions to the text and incorporated the suggestions provided by the reviewers.

This manuscript is a resubmission of an earlier submission. The following is a list of the peer review reports and author responses from that submission.

Round 1

Reviewer 1 Report

Comments and Suggestions for Authors

Authors stated that their simulation results are important to understand the ionization of crustal matter into the atmosphere, the vertical migration of charged particles, and ultimately the formation of vertical electrical separation in the air, resulting in additional vertical polarization electric field.

As noted, communications are short articles that present groundbreaking preliminary results or significant findings that are part of a larger study over multiple years (cutting-edge methods/ experiments, new technology or materials.).

Accordingly, I think it is not enough and convincing that on the basis of simulations without much discussion and analysis the authors can claim/prove such findings. I would recommend that the authors expand and improve the study and also change the submission type to article. I also recommend that they put some of the software and studies on github or similar to make it available to everyone.

Author Response

# Reviewer 1

Authors stated that their simulation results are important to understand the ionization of crustal matter into the atmosphere, the vertical migration of charged particles, and ultimately the formation of vertical electrical separation in the air, resulting in additional vertical polarization electric field.

As noted, communications are short articles that present groundbreaking preliminary results or significant findings that are part of a larger study over multiple years (cutting-edge methods/ experiments, new technology or materials.).

Accordingly, I think it is not enough and convincing that on the basis of simulations without much discussion and analysis the authors can claim/prove such findings. I would recommend that the authors expand and improve the study and also change the submission type to article. I also recommend that they put some of the software and studies on github or similar to make it available to everyone.  

Response: [Thank you for your useful comments and suggestions. As you've described, communications are short articles that present groundbreaking preliminary results or significant findings, such a simulation work is the first time that it has been proposed and presented. Based on the length of this manuscript, we preferred the type of this paper to be communication. Then this paper has been expanded and improved in the revised version. We have put the source code on the github site and you can see on the website: https://github.com/leibupang123/particlemove.]

We tried our best to improve the manuscript and made some changes in the manuscript. We earnestly appreciate the Editors/Reviewers’ work and hope that the corrections will be met with approval. Once again, thank you very much for your comments and suggestions.

Reviewer 2 Report

Comments and Suggestions for Authors

The introduction has to discuss more extensively the inputs coming from the cited references.

The physical modelling must be discussed and explained in deeper details.

Also the simulation has to be described better, at this stage it is not clear at all.

The figure captions must be more self-explanatory.

Also the figures itself have to be clearer presented (for instance as an example figure 2).

The references must be presented in a temporal order, strictly following the format of the journal.

Finally, last but not least, I suggest to separate the discussion from the conclusions. The two paragraphs must adhere to the line of the paper, presenting in a clearer way the results.    

Comments on the Quality of English Language

The level of English is not appropriate, the suggestion is to revise extensively the text by an English mother-tongue with a scientific background

Author Response

# Reviewer 2

The introduction has to discuss more extensively the inputs coming from the cited references.

The physical modelling must be discussed and explained in deeper details.

Also the simulation has to be described better, at this stage it is not clear at all.

Response: [Thank you and the introduction, the physical modelling and simulation have been discussed in deeper details in the revised manuscript.]

The figure captions must be more self-explanatory.

Also the figures itself have to be clearer presented (for instance as an example figure 2).

Response: [Thank you, the figure captions and the figures have been improved.]

The references must be presented in a temporal order, strictly following the format of the journal.

Response: [The references have been improved.]

Finally, last but not least, I suggest to separate the discussion from the conclusions. The two paragraphs must adhere to the line of the paper, presenting in a clearer way the results. 

Response: [Thank you, the discussion has been separated from the conclusions, and these two parts have been improved in the revised version.]

We tried our best to improve the manuscript and made some changes in the manuscript. We earnestly appreciate the Editors/Reviewers’ work and hope that the corrections will be met with approval. Once again, thank you very much for your comments and suggestions.

Reviewer 3 Report

Comments and Suggestions for Authors

The scientific steps of the research were not correctly followed. The experimental results have not confirmed the explanations provided. The manuscript does not contain enough information to be published as an article.

Comments on the Quality of English Language

The manuscript's English language is weak and contains grammatical and editing errors.

Author Response

# Reviewer 3

The scientific steps of the research were not correctly followed. The experimental results have not confirmed the explanations provided. The manuscript does not contain enough information to be published as an article.

Response: [Thank you for your comments, the introduction, scientific steps of the research and experimental results have been improved in the revised version.]

We tried our best to improve the manuscript and made some changes in the manuscript. We earnestly appreciate the Editors/Reviewers’ work and hope that the corrections will be met with approval. Once again, thank you very much for your comments and suggestions.

Round 2

Reviewer 1 Report

Comments and Suggestions for Authors

The authors have attempted to revise the text.
Although the authors expanded the text and put the software on github
they didn't expand and improve the study what is crucial.
I think it is not enough and I cant recommend the text for accepting.

Author Response

# Reviewer

The authors have attempted to revise the text.
Although the authors expanded the text and put the software on github

they didn't expand and improve the study what is crucial.
I think it is not enough and I cant recommend the text for accepting. 

Response: [Thank you for your useful comments and some suggestions. As you've concerned, our previous simulation process was not considered in enough detail, and the variability of thermal convective forces over time has not been taken into account. Assuming that F is a exponential decreasing quantity with time, We have optimized this model in our revised manuscript, the simulation results as shown in Figure 3.]

We tried our best to improve the manuscript and made some changes in the manuscript. We earnestly appreciate the Editors/Reviewers’ work and hope that the corrections will be met with approval. Once again, thank you very much for your comments and suggestions.
